# Genomic Markers of CDK 4/6 Inhibitor Resistance in Hormone Receptor Positive Metastatic Breast Cancer

**DOI:** 10.3390/cancers14133159

**Published:** 2022-06-28

**Authors:** Jin Sun Lee, Susan E. Yost, Sierra Min Li, Yujie Cui, Paul H. Frankel, Yate-Ching Yuan, Daniel Schmolze, Colt A. Egelston, Weihua Guo, Mireya Murga, Helen Chang, Linda Bosserman, Yuan Yuan

**Affiliations:** 1Department of Medical Oncology & Therapeutics Research, City of Hope Comprehensive Cancer Center, Duarte, CA 91010, USA; jin.bitar@gmail.com (J.S.L.); suyost@coh.org (S.E.Y.); mmurga@coh.org (M.M.); helchang@coh.org (H.C.); lbosserman@coh.org (L.B.); 2Department of Biostatistics, City of Hope Comprehensive Cancer Center, Duarte, CA 91010, USA; mli@coh.org (S.M.L.); yucui@coh.org (Y.C.); pfrankel@coh.org (P.H.F.); 3Department of Computational Quantitative Medicine, City of Hope Comprehensive Cancer Center, Duarte, CA 91010, USA; yyuan@coh.org; 4Department of Pathology, City of Hope Comprehensive Cancer Center, Duarte, CA 91010, USA; dschmolze@coh.org; 5Department of Immuno-Oncology, City of Hope Comprehensive Cancer Center, Duarte, CA 91010, USA; cegelston@coh.org (C.A.E.); wguo@coh.org (W.G.)

**Keywords:** CDK 4/6 inhibitor, genomic marker of resistance, metastatic breast cancer

## Abstract

**Simple Summary:**

The mechanisms of intrinsic and acquired resistance to CDK4/6 inhibitors are poorly understood. Patients with HR+ MBC treated with CDK4/6i and antiestrogen therapy were grouped into early (<6 months), intermediate (6–24 months for 0–1 lines; 6–9 months for ≥2 lines), or late progressors (>24 months for 0–1 lines; >9 months PFS for ≥2 lines). Among the 109 patients who received CDK4/6i as 1st- or 2nd-line therapy, 17 genes showed association with PFS (*p*-value < 0.15 and HR ≥ 1.5 or HR < 0.5). RNAseq was analyzed for 24/109 (22%) patients with 0–1 prior lines of therapy and 56 genes associated with PFS (HR ≥ 4 or HR ≤ 0.25 and FDR ≤ 0.15). Genomic biomarkers including *FGFR1* amplification, *PTEN* loss, and DNA repair pathway gene mutations showed significant associations with shorter PFS for patients receiving CDK4/6 inhibitor therapy.

**Abstract:**

Cyclin-dependent kinase 4/6 inhibitors are the standard of care for hormone receptor-positive metastatic breast cancer. This retrospective study reports on genomic biomarkers of CDK 4/6i resistance utilizing genomic data acquired through routine clinical practice. Patients with HR+ MBC treated with palbociclib, ribociclib, or abemaciclib and antiestrogen therapy were identified. Patients were grouped into early (<6 months); intermediate (6–24 months for 0–1 lines; 6–9 months for ≥2 lines); or late progressors (>24 months for 0–1 lines; >9 months PFS for ≥2 lines). NGS and RNA sequencing data were analyzed in association with PFS, and survival analysis was stratified by prior lines of chemotherapy. A total of 795 patients with HR+ MBC treated with CDK 4/6i were identified. Of these, 144 (18%) patients had genomic data and 29 (3.6%) had RNA data. Among the 109 patients who received CDK4/6i as 1st- or 2nd-line therapy, 17 genes showed associations with PFS (*p*-value ≤ 0.15 and HR ≥ 1.5 or HR < 0.5). Whole transcriptome RNAseq was analyzed for 24/109 (22%) patients with 0–1 prior lines of therapy and 56 genes associated with PFS (HR ≥ 4 or HR ≤ 0.25 and FDR ≤ 0.15). In this retrospective analysis, genomic biomarkers including *FGFR1* amplification, *PTEN* loss, and DNA repair pathway gene mutations showed significant associations with shorter PFS for patients receiving CDK4/6 inhibitor therapy.

## 1. Introduction

Cyclin-dependent kinase 4/6 inhibitors (CDK4/6i) induce G1 cell-cycle arrest and have shown significant survival benefits when combined with hormonal therapy in the treatment of hormonal receptor positive (HR+) metastatic breast cancer (MBC) [1]. The three FDA-approved CDK4/6i, palbociclib, ribociclib, and abemaciclib, have shown progression-free survival (PFS) benefits [2,3,4,5,6,7]. Updated results of MONARCH 2, MONALEESA 3 and MONALEESA 7 trials further confirmed the overall survival (OS) benefits of abemaciclib and ribociclib [8,9,10,11]. However, despite the significant survival benefits, tumor progression is inevitable for most patients.

Both intrinsic and acquired CDK4/6i resistance have been described [12,13,14], including cell cycle-dependent mechanisms such as *RB* loss [15,16], *CDK4* or *CDK6* amplification [17], *CDK7* or cyclin D overexpression, *WEE1* overexpression, cyclin E-CDK2 activation [18,19], loss of CDK repressors, *MDM2* overexpression, *HDAC* activation, and loss of *FZR1* [20]. Other non-cell-cycle-dependent mechanisms have also been described, including the activation of the PI3K/AKT/mTOR signaling pathway, FGFR pathway activation, the loss of ER expression, and *ESR1* mutation [20,21,22]. These potential biomarkers were tested using tumor specimens collected from pivotal clinical trials; however, the results were controversial. ER, RB, p16, cyclin D1, and Ki-67 were tested in the PALOMA-2 trial and were not associated with sensitivity or resistance to palbociclib [23]. In the PALOMA-3 trial, palbociclib was beneficial regardless of mutations in *ESR1*, and *PIK3CA* mutations predicted sensitivity to palbociclib [24].

The mechanisms of CDK4/6i resistance are poorly understood in the real-world setting. Broad-panel next-generation sequencing (NGS) has become widely available in routine clinical practice and offers the ability to identify potentially actionable genomic alterations. The utility of the broad panel NGS is yet to be defined for patients with MBC [25,26]. Therefore, the current retrospective analysis was initiated to understand the molecular mechanism of resistance to CDK4/6i by analyzing tumor genomic data acquired through routine clinical care in patients who received CDK4/6i as standard-of-care therapy.

## 2. Materials and Methods

### 2.1. Patients

This study was conducted using an institutional review board (IRB)-approved protocol. Patients with HR+ MBC who had tumor NGS testing and were treated with at least one CDK4/6i (palbociclib, ribociclib, or abemaciclib) and antiestrogen therapy between January 2015 and July 2020 were identified through electronic medical record (EMR) search. Patient characteristics and disease variables were obtained by chart review: age, race/ethnicity, histology type, initial tumor stage; prior lines of therapy for metastatic disease; systemic treatment for metastasis (CDK4/6i and endocrine therapy); and sites of metastases.

Response to therapy was evaluated through routine staging with CT or PET-CT, and PFS and OS were determined based on chart review. The results from germline genetic testing were collected if applicable. PFS on CDK4/6i was calculated from the first day of treatment until the date of disease progression. The response to CDK4/6i was classified as follows: early progressors (PFS ≤ 6 months), intermediate progressors (PFS 6–24 months for 1st line patients and 6–9 months for ≥2 lines of therapy), and late progressors (PFS ≥ 24 months for first line patients and ≥ 9 months for ≥2 lines of therapy)(21).

### 2.2. Tumor Genomics Analysis

Of 144 patients with genomic results, a total of 43 (30%) specimens were sequenced with the Tempus comprehensive xT^®^ platform, which includes a targeted 648 gene panel. Of those, 79 (55%) samples were sequenced using FoundationOne^®^ CDx comprehensive profiling, which includes 324 mutations. A total of 22 (15%) of these samples were sequenced using HopeSeq platform, including 89 “hot-spot” genes [27]. An additional 9 (6%) patient samples were sequenced using GEM ExTra^®^, which covers all protein coding regions [28]. To study the associations of PFS and genomic alterations, we evaluated the hazard ratio (HR) for each gene associated with PFS using a Cox proportional hazard model.

### 2.3. Expression Analysis

Of the 109 patients who received 0–1 prior lines of therapy, 24 (22%) had whole transcriptome RNAseq via Tempus^®^ [29]. The Cox proportional hazard model was used to evaluate the HR for PFS for 11,426 protein coding genes after filtering. The false discovery rate (FDR) was used to adjust for multiple comparisons. Only protein-coding genes were evaluated, and genes with low expression or variation, with average expression or standard deviation less than 25% of genome-wide quantile, were filtered.

### 2.4. Survival Analysis

PFS data were plotted as Kaplan–Meier curves. We compared PFS among patients with 0–1 prior line of therapy (n = 109) vs. ≥2 lines of therapy (n = 35) with a log-rank test. We further compared PFS among 0, 1, and ≥2 lines of therapy with n = 80, 29, and 35 patients, respectively. The same analyses were performed for overall survival (OS) with a log-rank test.

## 3. Results

### 3.1. Patients

A total of 795 patients with stage IV HR+ MBC who received CDK4/6i were identified. Of these, 696 (87%) patients received palbociclib, 60 (8%) patients received abemaciclib, and 39 (5%) patients received ribociclib from January 2015 to July 2020 at City of Hope; the data cut-off was April 2022. A total of 144 (18%) patients had commercial NGS test results. Of the 144 patients, 80 (56%) patients had received no prior line, 29 (20%) patients had received 1 prior line, and 35 (24%) patients had received 2+ lines of treatment for metastatic disease prior to starting CDK4/6i (Figure 1). Patients were grouped by PFS: Early progressors (PFS < 6 months; N = 41), intermediate progressors (PFS 6–24 months for 0–1 line; 6–9 months for 2+ lines; N = 66), and late progressors (PFS >24 months for 0–1 line; >9 months for 2+ lines; N = 37). Of the 144 patients with NGS results, 29 patients with Tempus^®^ testing had RNA seq data available. Of these, 24 patients had 0–1 line of prior therapy and were analyzed for differential gene expression.

Of the 144 patients included in the final analysis, the median age was 57 (range, 23–84) (Table 1). A total of 66 (46%) patients were non-Hispanic white, 42 (29%) were Hispanic, 24 (17%) were Asian, and 8 (6%) were African American. Histology type included 99 (69%) invasive ductal carcinoma (IDC), 28 (19%) invasive lobular carcinoma (ILC), 2 (1%) mixed IDC/ILC, and 15 (11%) tumors of other histology. At initial diagnosis, the different stages were 19 (13%) stage I, 40 (28%) stage II, 35 (24%) stage III, 39 (27%) de novo stage IV, and 11 (8%) unknown stage. Of the 144 patients, 104 (72%) had visceral metastasis and 40 (28%) had non-visceral metastasis at time of initiating CDK4/6 inhibitor.

### 3.2. Tumor Genomic Alterations Associated with PFS

Of 144 patients with genomic results, the following tests were used: Tempus xT^®^ (N = 43), FoundationOne^®^ (N = 79), HopeSeq^®^ (N = 22) and GEM ExTra^®^ (N = 9). Genomic tile plots are shown in Appendix A. Tumor genomic analysis of patients with 0 or 1 line of prior therapy (N = 109) is shown in Appendix A.The associations between genomic alterations and PFS were tested, and 17 genes showed associations with PFS with HR ≥ 1.5 and *p*-value ≤ 0.15. The genomic alterations that were present in 2 or more patients with 0–1 line of prior therapy (N = 109) and that were associated with impaired PFS were: *MLL3* (HR = 4.00, *p* = 0.01, N = 4), *ZNF703* (HR = 2.55, *p* = 0.01, N = 8), *FGFR1* (HR = 1.78, *p* = 0.02, N = 23), *CDKN2B* (HR = 3.27, *p* = 0.02, N = 4), *GPR124* (HR = 5.34, *p* = 0.02, N = 2), *CDK4* (HR = 2.75, *p* = 0.03, N = 5), *PALB2* (HR = 2.83, *p* = 0.05, N = 4), *MAP2K4* (HR = 2.02, *p* = 0.06, N = 8), *BRCA1* (HR = 3.93, *p* = 0.06, N = 2), *RB1* (HR = 2.62, *p* = 0.06, N = 4), *MDM2* (HR = 2.19, *p* = 0.06, N = 6), *CREBBP* (HR = 2.86, *p* = 0.08, N = 3), *FAT1* (HR = 0.30, *p* = 0.09, N = 2), *FRS2* (HR = 2.66, *p* = 0.1, N = 3), *ATRX* (HR = 3.17, *p* = 0.11, N = 2), *PTEN* (HR = 1.45, *p* = 0.12, N = 22), and *ERBB2* (HR = 1.76, *p* = 0.13, N = 8) (Table 2). The most common alterations associated with decreased PFS were *FGFR1* amplification (HR 1.78, *p* = 0.02), found in 23 (21%) patients, and *PTEN* loss (HR 1.45, *p* = 0.12), found in 22 (20%) patients.

### 3.3. mRNA Expression Associated with PFS

Of the 144 patients with genomic sequence data, 109 patients had 0–1 prior line of therapy, and 24/109 (22%) had whole transcriptome RNAseq via Tempus^®^ [29]. Coding gene fragments per kilobase of exon per million (FPKM) mapped fragments were analyzed in patients with continuous PFS (Figure 2). A total of 11,426 protein-coding genes were studied, and a false discovery rate (FDR) was calculated to adjust for multiple comparisons. The 56 genes in the heatmap were selected with HR ≥ 4 or HR ≤ 0.25 and FDR ≤ 0.15. Unsupervised hierarchical clustering was performed and identified 3 clusters with distinct gene expression patterns. The 3 clusters are characterized as early (N= 4), intermediate (N = 12), or late (N = 8) progressors, with median PFS of 4.5 (N = 4 range), 8.5 (N = 12 range), and 29 (N = 8, range) months, respectively. Statistical analysis was conducted in R (R-3.6.3, R Core Team) [30]. RNA analysis for patients with 0 or 1 line of prior therapy (N = 24) is shown in Appendix A

### 3.4. Survival

Kaplan-Meier survival analysis stratified by prior lines of chemotherapy showed that the median PFS for 0–1 prior line (N = 109) was 12 months (95% CI 10, 17) and for ≥2 lines (N = 35) was 7 months (95% CI 4, 12). Median OS for 0–1 prior line (N = 109) was 39 months (95% CI 33, 51), and for ≥2 lines (N = 35), it was 28 months (95% CI 21, 41) (Appendix A). To further analyze survival, patients were separated into 0, 1, and ≥2 prior lines of chemotherapy. The results showed median PFS for 0 prior lines (N = 80, red line) of 12.5 months (95% CI 9–19); median PFS for 1 prior line (N = 29, green line) of 12 months (95% CI 10–18); and median PFS for ≥2 lines (N = 35, blue line) of 7 months (95% CI 4–12). Median OS for 0 prior lines (N = 80, red line) was 37 months (95% CI 30–50); median PFS for 1 prior line (N = 29, green line) was 54 months (95% CI 39–NA); and median PFS for ≥2 lines (N = 35) was 28 months (95% CI 21–41) (Appendix A). Patients who did not progress were censored at the date of their last follow-up. The decreased PFS compared with the historic data may be explained by prior endocrine treatment for metastatic disease in this patient population: 93 (71%) patients had prior aromatase inhibitor (AI), 33 (25.2%) patients had prior fulvestrant, 2 (1.5%) patients had prior tamoxifen, and only 3 (2.3%) patients were endocrine therapy naive.

## 4. Discussion

CDK4/6i has revolutionized the treatment of advanced HR+ MBC, and the indication has been expanded to early-stage breast cancer with remarkable results from the monarchE trial [31,32]. However, the benefits are not observed across all patients due to intrinsic or acquired resistance. Understanding the mechanism of the resistance to CDK4/6i is critical for personalized treatment in breast cancer. Multiple studies were conducted to understand the molecular mechanisms of resistance to CDK 4/6i in breast cancer [33,34]. Paired baseline and end-of-treatment ctDNA from 195 patients in the PALOMA-3 was studied [16] and showed that acquired mutations (mut) in *RB1* occurred in 5% of patients after palbociclib. In addition, new driver mutations in *PIK3CA* and *ESR1* were observed. Pooled ctDNA analysis from the MONALEESA trial evaluating the efficacy of ribociclib plus different endocrine therapies demonstrated potential biomarkers associated with response, including F*RS2*, *MDM2*, *PRKCA*, *ERBB2*, *AKT1*, and *BRCA1/2*, and biomarkers associated with resistance, including *CHD4*, *BCL11B*, *ATM*, or *CDKN2A/2B/2C* [12]. Li et al. reported associations of *RB1* loss, *FAT1* loss (Hippo pathway) and de novo resistance to CDK4/6i using MSK-IMPACT [35]. Formisano et al. reported *FGFR1/ZNF703* amplification (amp) in 20/427 (4.7%) pre-treatment ctDNA samples associated with reduced clinical benefit from ribociclib in patients enrolled in the MONALEESA-2 trial [36]. Lastly, NGS on 59 tumors with CDK4/6i exposure revealed resistance mechanisms: *RB1* loss; activating alterations in *AKT1, RAS, AURKA, CCNE2, ERBB2*, and *FGFR2*; and the loss of ER expression [37]. In the current study, despite the limited sample size and the low frequency of many somatic mutations, several statistically significant patterns were identified, including genomic alterations associated with CDK4/6i resistance: *FGFR1* amp, *ZNF703* amp, *CDK4* amp, *PALB2* mut, *BRCA1* mut, *RB1* loss, *CDKN2B* loss, *CREBBP* mut, *MLL3* mut, *ERBB2* mut, *PTEN* loss, *MAP2K4* mut, *MDM2* amp, *FRS2* amp, *ATRX* mut, and *GPR124* mut. In addition, *FAT1* mut/loss was associated with increased PFS. In this retrospective analysis using real-world data, molecular biomarkers such as *FGFR1* amplification, *PTEN* loss, and DNA repair pathway gene mutations showed significant association with shorter PFS with CDK4/6i therapy.

CDK 4 and 6 control cellular transition from the G1 phase to S phase of the cell cycle and activated cyclin D and CDK4/6 complex promotes the initiation of DNA synthesis and entry into S-phase by retinoblastoma phosphorylation that de-represses activity of E2F family facilitating cyclin E1 and E2 activity. Cyclin E then activates *CDK2*, which hyper-phosphorylates RB, further increasing the expression of *E2F* target genes that are critical for the cell to proceed into S phase. The loss or mutation of *RB*, the main target of CDK4/6, is a well-known driver of the resistance. Condorelli et al. reported acquired somatic *RB1* mutations from 3 patients with MBC after exposure to CDK4/6i [15]. The *RB1* mutations were identified at 5, 8, and 13 months after initiation of CDK4/6i in each patient from circulating tumor DNA (ctDNA) analyzed by commercially available NGS-based assay. The biomarker study performed in the PALOMA-3 trial evaluating the efficacy of fulvestrant plus palbociclib demonstrated that Cyclin E1 (*CCNE1*) expression was associated with resistance to Palbociclib, and low expression of *CCNE1* mRNA was related to better efficacy of palbociclib [18]. Our finding of reduced PFS in association with *CDK4* amp, *RB1* loss, and *CDKN2B* loss is consistent with the above findings. *CDKN2A* and *CDKN2B* are endogenous inhibitors of CDK 4/6. High *CDKN2* levels may predict reduced CDK4/6 activity and resistance to CDK4/6i [34,38]. Due to our limited sample size, our finding of *CDKN2B* loss in association with shorter PFS needs to be further verified. Hippo signaling in ER+ breast cancer is an established tumor suppressor in ER+ MBC, and *FAT1* loss serves as a mechanism of resistance to CDK4/6i [35]. Our finding of *FAT1* mutation/loss in association with better PFS could be due to the small sample size.

Several common, potentially actionable resistance mechanisms involving growth factor pathways have been identified, including the FGF, Aurora Kinase, AKT, and MAPK signaling pathways [39]. The fibroblast growth factor receptor (FGFR) pathway is involved in cell proliferation, differentiation, and growth and plays a role in CDK4/6i therapy resistance. *FGFR1* amplification has been shown to trigger the recruitment and activation of *STAT3* [40] and may be involved in multiple effector pathways that activate ERK1/2 in different cellular contexts (“additive” signaling) [41]. ctDNA analysis from patients enrolled in MONALEESA-2(3) demonstrated shorter PFS in patients with *FGFR1* amplification compared with patients with wild-type *FGFR1* [36,42]. The ablation of *PTEN*, through increased *AKT* activation, was sufficient to promote resistance to CDK4/6 inhibition in breast cancer cells in vitro and in vivo [14]. *PTEN* loss resulted in the exclusion of p27 from the nucleus, leading to the increased activation of both *CDK4* and *CDK2* [14]. Our finding of *PTEN* loss and the trend of reduced PFS is consistent with the above findings. *ZNF703* amplification, predominantly identified in Luminal B subtype of breast cancer, is associated with poor clinical outcomes [43]. In the ctDNA analysis of the MONALEESA-2 trial, Formisano et al. identified that5% (20/427 patients) patients had copy number alterations of the 8p11.23 genomic locus, which harbors *FGFR1* and *ZNF703* genes and is associated with reduced clinical benefit for ribociclib (median PFS 10.61 vs. 24.84 months, *p* = 0.075) [36]. *ERBB2* mutations were identified in 5/41 CDK4/6-resistant tumor [37], and preclinical work demonstrated that *ERBB2* mutation activates downstream MAPK/AKT/mTOR and confers resistance to CDK4/6 blockade in breast cancer cells [44]. *MAP2K4* encodes a member of the MAPK family, and *MAP2K4* mutation was identified in breast cancer resistant to AI [45]. Fibroblast growth factor receptor substrate 2 (*FRS2*) acts upstream of the FGF signaling pathway, is an adaptor protein expressed in a small subset of epithelial cells and triggers a cytokine-rich inflammatory microenvironment that promotes breast cancer carcinogenesis [46].

DNA repair defect (DRD) gene mutations may confer resistance to CDK4/6i. Safonov et al. recently reported *gBRCA2* mutation in association with significantly inferior PFS (HR 2.17, 95% CI 1.46–3.22, *p* < 0.001) for 1st-line treatment with CDK4/6 I [47]. No significant association between *PALB2* mutation and PFS with CDK4/6i was identified in this study. In a real-world study of CDK4/6i including 2698 patients, 9.9% had *gBRCA2* mutation, which is associated with shorter time to subsequent therapy or death (HR1.24, 95% CI 0.96–1.59) [48]. In our study, *gBRCA1* mutation was reported in 2/144 (1.4%) patients and was associated with a trend of inferior PFS (HR3.93, 95% CI 0.94–16.5, *p* = 0.06). 7/144 (5%) patients had *gBRCA2* mutation, but no association with PFS was observed (HR 1.34, 95% CI 0.62–2.90, *p* = 0.46). *MDM2* and *MDM4* inhibit *TP53*, and amplification renders resistance in ER+ BC patients. *MDM2* inhibits DNA break repair through association with the Mre11/Rad50/Nbs1 DNA repair complex [49]. Approximately 30–50% of HR+ MBC showed mutation of *P53* or regulators *MDM2* and *MDM4* [37]. *MDM2* inhibitor could provide a therapeutic opportunity, particularly for those malignancies that have lost functional p53. Our finding of *MDM2* mutation associated with CDK4/6i resistance is consistent with this finding. Alpha Thalassemia/Mental Retardation Syndrome X-Linked (*ATRX*) loss is frequently identified in gliomas and may be a molecular marker for DNA damage response defects and *ATRX* knock-out led to PARP inhibitor sensitivity in glioma cells [50]. The role of *ATRX* in breast cancer remains unclear.

Mixed lineage leukemia gene (*MLL3*), a histone monomethylase that is known to interact with nuclear hormone receptors such as ERα, is frequently mutated in multiple cancer types. *MLL3* is the sixth most mutated gene in ER+ breast cancer: 8.5% in AACR GENIE [51] and 9% of ER+ breast cancer patients in TCGA [52]. *MLL3* mutation is a de novo cause of endocrine therapy resistance in ER+ breast cancer and the mutation pattern of *MLL3* in breast cancer is most consistent with a haploinsufficient tumor suppressor [53]. G-protein coupled receptor 124 (*GPR124*) is significantly expressed in tumor vasculature and play a critical role in VEGR-induced tumor angiogenesis, which involves cell-cell interaction, permeability, migration, and invasion [54]. *CREBBP* binds to CREB, which is known to play a critical role in embryonic development. CREBBP/EP300 HAT inhibition suppresses EGR-dependent transcription in breast cancer in vitro and in vivo [55].

Analyzing mRNA expression, Turner et al. reported that high *CCNE1* mRNA expression predicted worse response to palbociclib in the PALOM-3 trial. High *CCNE1*, *CCND3*, and *CDKN2B* mRNA expression levels were associated with resistance to palbociclib in the NeoPalAna trial(18). Guerrero-Zotano et al. reported that high E2F4 signature activity correlated with response to palbociclib [56]. In the current study, mRNA expression predicting worse response to CDK4/6i included increased expression of *PARP1* (average expression 1.98, HR 6.30 and *p* = 0.00007) and *STAT1* (average expression 0.94, HR 4.42, and *p* = 0.002), and decreased expression of *BCROL1* (average expression −0.39, HR 0.21, *p* = 0.001) and *NFRKB* (average expression 0.58, HR 0.12, *p* = 0.002). Due to the limited sample size, these findings need to be further verified through larger studies.

In our study, 16% of patients underwent tumor genomic testing. The genomic tests were utilized significantly less frequently in community network sites compared with the main campus (7% vs. 18%). The patients included in this study were treated prior to 2020, when broad-panel NGS was commonly adopted as routine clinical practice. NCCN Guidelines for Breast Cancer V.1.2021 include NGS and comprehensive genomic profiling (CGP) as a method of detecting actionable mutations and fusions such as *BRCA1/2* mutations, *PIK3CA* mutations, *ESR1* mutation, *HER2* mutation, MSI-H status, and deficient mismatch repair (dMMR) [57]. Integrating community oncologists into the academic paradigm of personalized medicine, including continued development of interpersonal relationships between community oncologists and academic site physicians through molecular tumor board teams, will play a major role in personalized therapy advancements in oncology [58].

There were several limitations to our study including a limited sample size, with only a fraction of patients receiving CDK4/6i genomic sequencing, which limits the statistical power of the current analysis. The recently updated NCCN 2021 guidelines recommend routine NGS testing for patients with metastatic breast cancer, which likely will increase utilization in community practice. Since this study was retrospective in nature, the genomic tests were performed through a variety of platforms per the oncologists’ preference. The adoption of institutional practice guidelines will also facilitate an increased number of patients receiving genomic sequencing, the appropriate utilization of repeat biopsy upon disease progression, the increased availability of adequate tissue specimens for NGS, and the minimized inter-test variance by minimizing the variation among different next-generation sequencing platforms. Currently, an institutional effort is underway to optimize a precision medicine approach and mitigate the aforementioned challenges. It is also noted that the median PFS in this study for patient who received 0–1 line of prior therapy (N = 109) appears to be shorter than that in the real-world study reported by DeMichele et al. (N = 1430) [59]. Prior endocrine therapy and the relatively high percentage (72%) of patients with visceral metastasis may explain the shorter PFS in our study.

## 5. Conclusions

In this retrospective analysis, genomic biomarkers such as *FGFR1* amplification, *PTEN* loss, and DNA repair pathway gene mutations showed significant associations with shorter PFS for patients receiving CDK4/6 inhibitor therapy.

## Figures and Tables

**Figure 1 cancers-14-03159-f001:**
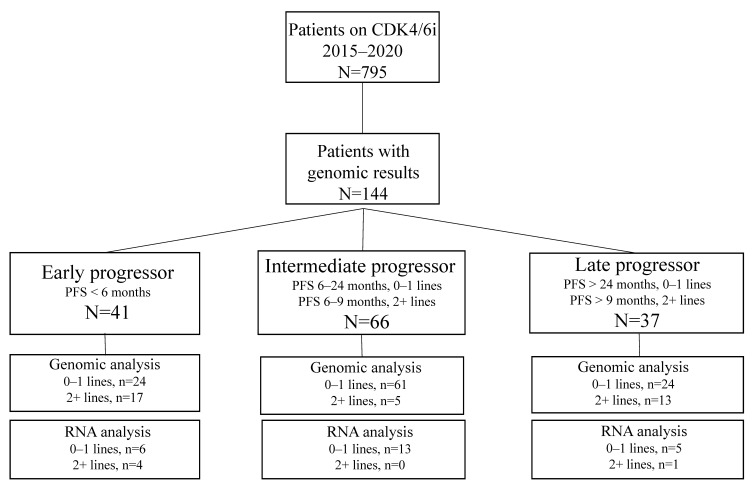
Flow chart showing patients included in the study and the availability of NGS and RNAseq results. Patients are grouped by PFS: Early progressors (PFS < 6 months; N = 41), intermediate progressors (PFS 6–24 months for 0–1 line; 6–9 months for 2+ lines; N = 66), and late progressors (PFS > 24 months for 0–1 line; >9 months for 2+ lines; N = 37). Of 144 patients with NGS results, 29 patients underwent Tempus testing and had RNA seq data available (N = 24 patients with 0–1 line were used for differential gene expression).

**Figure 2 cancers-14-03159-f002:**
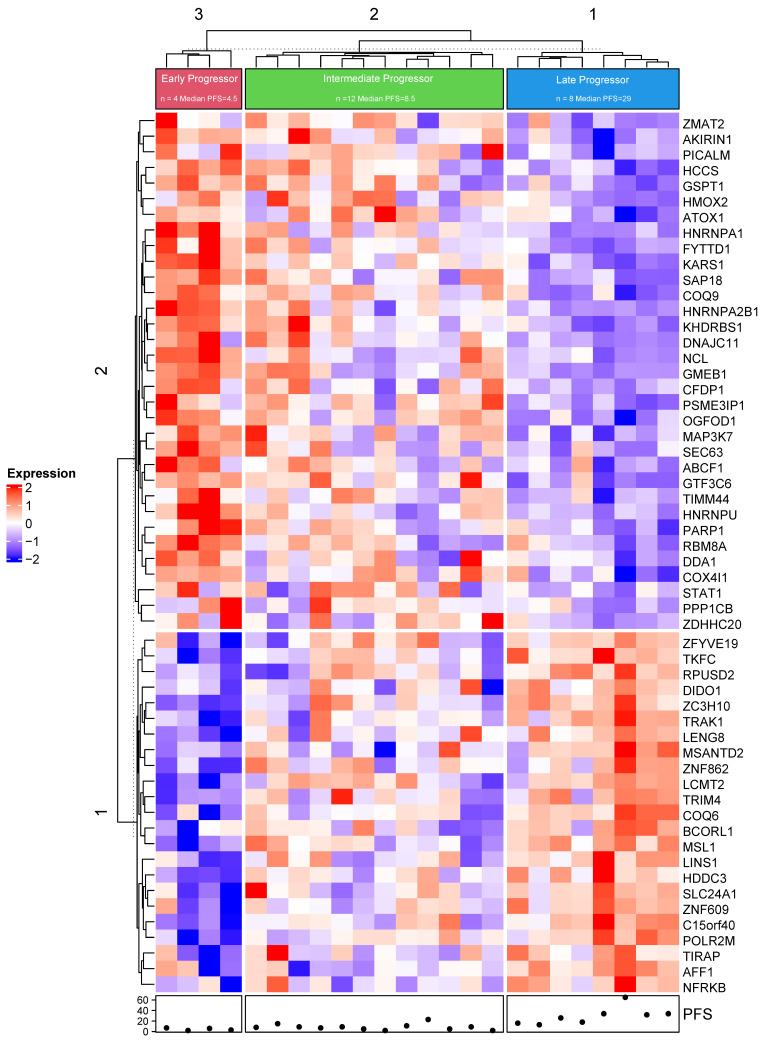
Differential gene expression. Whole transcriptome RNAseq was available for 24 (22%) patients with 0–1 prior line of therapy, and 56 genes were selected with HR ≥ 4 or HR ≤ 0.25 and FDR ≤ 0.15. The differential gene expression of all coding gene fragments per kilobase of exon per million (FPKM) mapped fragments were mapped in patients with continuous progression-free survival (N = 29) (cluster 3 (N = 4) median PFS = 4.5 months; cluster 2 (N = 12) median PFS = 8.5 months; and cluster 1 (N = 8) median PFS = 29 months). PFS is shown below the heat map (range 1, 65).

**Table 1 cancers-14-03159-t001:** Patient characteristics and treatment variables.

Characteristic	Patients Receiving CDK4/6i (N = 144)
Age, median (range) years	57 (23–84)
Race, N (%)	
White	109 (76%)
Asian	25 (17%)
African American	8 (6%)
Unknown	2 (1%)
Ethnicity, N (%)	
Hispanic	42 (29%)
Non-Hispanic	100 (70%)
Unknown	2 (1%)
Histology Type, N (%)	
IDC	99 (69%)
ILC	28 (19%)
IDC/ILC	2 (1%)
Others ^1^	15 (11%)
Tumor stage at initial diagnosis, N (%)	
I	22 (15%)
II	40 (28%)
III	33 (23%)
IV	49 (34%)
Number of prior lines, N (%)	
0	80 (56%)
1	29 (20%)
2+	35 (24%)
Sites of metastases, N (%)	
Visceral (liver, lung, CNS)	104 (72%)
Non-visceral (bone, LN, skin)	40 (28%)

^1^ Intraductal papillary (n = 3), metaplastic (n = 1), unknown (n = 11).

**Table 2 cancers-14-03159-t002:** Genomic alterations associated with PFS in patients who received CDK4/6i as 1st- or 2nd-line therapy (N = 109): 17 genes were significantly associated with impaired PFS in the Cox proportional hazard model with *p*-value ≤ 0.15 and HR ≥ 1.5 or HR ≤ 0.5.

	Patients (n)	HR	*p*-Value	95% CI	FDR
*MLL3* mut	4	4.01	0.01	1.42–11.28	0.32
*ZNF703* amp	8	2.55	0.01	1.21–5.378	0.32
*FGFR1* amp	23	1.78	0.02	1.09–2.90	0.32
*CDKN2B* loss	4	3.27	0.02	1.18–9.11	0.32
*GPR124* mut	2	5.34	0.02	1.25–22.86	0.32
*CDK4* amp	5	2.75	0.03	1.09–6.93	0.36
*PALB2* mut	4	2.83	0.05	1.01–7.90	0.40
*MAP2K4* mut	8	2.02	0.06	0.986–4.19	0.40
*BRCA1* mut	2	3.93	0.06	0.94–16.5	0.40
*RB1* loss	4	2.62	0.06	0.95–7.25	0.40
*MDM2* amp	6	2.19	0.06	0.95–5.05	0.40
*CREBBP* mut	3	2.86	0.08	0.90–9.13	0.43
*FAT1* mut/loss	2	0.30	0.09	0.07–1.22	0.52
*FRS2* amp	3	2.66	0.10	0.83–8.53	0.48
*ATRX* mut	2	3.17	0.11	0.77–13.05	0.50
*PTEN* loss	22	1.45	0.12	0.90–2.33	0.52
*ERBB2* mut	8	1.76	0.13	0.85–3.68	0.52

Mut, mutation; amp, amplification.

## Data Availability

Data supporting the reported results can be found at 10.5281/zenodo.6599679.

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
