# Peer review of "Genomic Markers of CDK 4/6 Inhibitor Resistance in Hormone Receptor Positive Metastatic Breast Cancer"

_cancers, 2022, doi:10.3390/cancers14133159_

Round 1
Reviewer 1 Report
This manuscript reports genomic biomarkers of CDK 4/6i resistance utilizing genomic data acquired through routine clinical practice. Patients with HR+ MBC treated with palbociclib, ribociclib, or abemaciclib and antiestrogen therapy were identified and were grouped into early, intermediate, and late progressors. According to analysis, genomic biomarkers including FGFR1 amplification, PTEN loss, and DNA repair pathway gene mutations showed significant association with shorter PFS for patients receiving CDK4/6 inhibitor therapy.
In conclusion, this manuscript contributes to understanding the molecular mechanism of resistance to CDK4/6i by analyzing tumor genomic data acquired through routine clinical care in patients who received CDK4/6i as standard-of-care therapy. For these reasons, the manuscript matches the criteria for publication in Cancers. However, one minor revision that needs to be fixed is necessary
-In Figure 2, the gene names are not clear, please make a more clear version.
Author Response
We thank the reviewer for their thoughtful comments. Figure 2 has been updated to make the gene names larger. We agree with the reviewer that there are limitations to our retrospective study including small sample size, but we believe the methods are adequately described and conclusions are supported by results in this single institution real-world study.
Reviewer 2 Report
Cyclin-dependent kinase 4 and 6 (CDK4/6) inhibitors have improved the treatment of hormone-positive metastatic breast cancers. Number of studies have been performed using next generation sequencing and protein analysis to understand the mechanism of CDK4/6 inhibitors and to identify the biomarkers for sensitivity or resistance to CDK4/6 inhibitors. In this article, the authors analyzed the resistance to CDK4/6 inhibitors by looking at the data set from patients who had received CDK4/6 inhibitors as part of their treatment. The findings mentioned in this current article regarding reduced PFS, RB1 loss, and CDKN2B loss are consistent with previous literature. In addition, the authors mention that genomic biomarkers like DNA repair pathway gene mutation and PTEN loss is associated with shorter PFS in patients receiving CDK4/6 inhibitor therapy. Overall, the new factors were not identified but using the dataset of patients receiving CDK4/6 inhibitor was very thoughtful.
Author Response
We thank the reviewer for their supportive and positive review!